# Copeptin Release in Arterial Hypotension and Its Association with Severity of Disease in Critically Ill Children

**DOI:** 10.3390/children9060794

**Published:** 2022-05-28

**Authors:** Philipp Baumann, Verena Gotta, Andrew Atkinson, Markus Deisenberg, Martin Hersberger, Adam Roggia, Kevin Schmid, Vincenzo Cannizzaro

**Affiliations:** 1Department of Intensive Care and Neonatology, University Children’s Hospital Zurich, University of Zurich, 8032 Zurich, Switzerland; markus.deisenberg@kispi.uzh.ch (M.D.); adamroggia@gmail.com (A.R.); kevin.schmid@kispi.uzh.ch (K.S.); 2Children’s Research Centre, University Children’s Hospital Zurich, University of Zurich, 8032 Zurich, Switzerland; martin.hersberger@kispi.uzh.ch (M.H.); vincenzo.cannizzaro@usz.ch (V.C.); 3Department of Paediatric Pharmacology and Pharmacometrics, University of Basel Children’s Hospital, 4056 Basel, Switzerland; verena.gotta@ukbb.ch (V.G.); andrew.atkinson@ukbb.ch (A.A.); 4Department of Anaesthesia, University Children’s Hospital Zurich, University of Zurich, 8032 Zurich, Switzerland; 5Clinical Chemistry and Biochemistry, University Children’s Hospital Zurich, University of Zurich, 8032 Zurich, Switzerland; 6Department of Neonatology, University Hospital Zurich, University of Zurich, 8091 Zurich, Switzerland

**Keywords:** neonate, intensive care, biomarker, antidiuretic hormone, vasopressin, arginine-vasopressin, vasopressor, inotrope, cardiopulmonary bypass

## Abstract

Low copeptin levels may indicate inadequate arginine-vasopressin release promoting arterial hypotension, whereas high copeptin concentrations may reflect disease severity. This single-center prospective non-randomized clinical trial analyzed the course of blood copeptin in critically ill normo- and hypotensive children and its association with disease severity. In 164 patients (median age 0.5 years (interquartile range 0.1, 2.9)), the mean copeptin concentration at baseline was 43.5 pmol/L. Though not significantly different after 61 h (primary outcome, mean individual change: −12%, *p* = 0.36, paired *t*-test), we detected 1.47-fold higher copeptin concentrations during arterial hypotension when compared to normotension (mixed-effect ANOVA, *p* = 0.01). In total, 8 out of 34 patients (23.5%) with low copeptin concentrations <10 pmol/L were hypotensive. Copeptin was highest in the adjusted mixed-effect regression analysis within the first day (+20% at 14 h) and decreased significantly at 108 h (−27%) compared to baseline (*p* = 0.002). Moreover, we found a significant association with vasopressor-inotrope treatment intensity, infancy (1–12 months) and cardiopulmonary bypass (all *p* ≤ 0.001). In conclusion, high copeptin values were associated with arterial hypotension and severity of disease in critically ill children. This study does not support the hypothesis that low copeptin values might be indicative of arginine-vasopressin deficiency.

## 1. Introduction

Critically ill children frequently suffer severe arterial hypotension in various conditions, including respiratory and cardiac failure, septic shock, postsurgery, asphyxia and brain injury. Treatment for arterial hypotension comprises intravenous fluid, vasopressor and inotrope administration to maintain adequate perfusion pressure [1,2,3]. However, excessive fluid administration may lead to volume overload, pulmonary edema and heart failure [4,5]. Application of vasoactive and inotropic drugs reduces the risk of fluid overload and is very efficient in improving perfusion pressure, but it carries the risks of arrhythmia, increased left and right ventricular afterload and diminished tissue blood flow below the hypoxic threshold [6,7,8].

Recently, intravenous vasopressin was introduced to the medical armament of pediatric intensive care medicine. It is used in addition to or instead of standard inotrope-vasopressor treatment [9,10,11,12]. Vasopressin is a natural endogenous hormone with probably fewer side effects than catecholamines. It lacks inotropic and chronotropic properties, saving the myocardium from increased oxygen consumption. Vasopressin has strong vasoconstrictive effects that are mediated by the V1-vascular arginine-vasopressin (V1-AVP) receptor [13]. Further, it plays a key role in endogenous water and electrolyte homeostasis in the collecting duct, as it induces intracellular cyclic adenosine monophosphate (cAMP) production and translocation of aquaporins from cytoplasmic vesicles to the apical cell membrane through binding to renal V2-AVP receptors [14]. Intravenous arginine-vasopressin administration in critically ill children seems to be safe [11,15,16] and most effective when endogenous AVP secretion is inadequately low [10,12,17,18,19].

Since endogenous plasma AVP is difficult to measure in vivo [20,21,22], the stable equimolarly released glycopeptide copeptin could help identify critically ill patients who might benefit directly from intravenous AVP administration. Copeptin is the 39-amino acid long C-terminal portion (C-terminal proAVP, CTproAVP) of the vasopressin precursor pre-provasopressin. It is released equimolarly in a 1:1 ratio to vasopressin into the portal circulation of the neurohypophysis. The function in the human body is unknown. Copeptin is better suited for laboratory assessment than vasopressin, as its concentrations in blood samples remain constant at room temperature for 7 days and up to 14 days at 4 °C [23]. Copeptin has been proposed as a potential surrogate biomarker for AVP deficiency but has not been systematically investigated in patients with arterial hypotension [24].

Further, there is a growing body of literature for the association of plasma copeptin with severity of disease and adverse outcome in both adults and children. It has been shown that non-favorable outcomes could be predicted in a variety of pediatric pathologies, such as community-acquired pneumonia [25,26], congenital heart disease with pulmonary hypertension [27] or perinatal asphyxia [28]. However, longitudinal data on pediatric patients in intensive care settings are still sparse [26,29,30,31].

The aim of this prospective observational study was to characterize the course of blood copeptin in critically ill children admitted to the intensive care unit (ICU). The primary objective was to describe the average change of blood copeptin concentrations in normotensive and hypotensive patients after admission to ICU. The secondary objective was to describe the change of blood copeptin over a period of up to 168 h and to explore the association of blood copeptin concentrations with the following clinical covariates: (a) age; (b) sex; (c) length of ICU stay; (d) length of respiratory support; (e) intensity of catecholamine therapy; (f) severity of disease score; and (g) rate of death at day 28. We first hypothesized an association between copeptin values and arterial hypotension, and second, a correlation between elevated copeptin concentrations and both severity of disease and adverse outcome.

## 2. Materials and Methods

This study was designed as a single-center prospective non-randomized clinical trial. It was conducted in critically ill children admitted to the mixed medical and surgical tertiary care intensive care unit of the University Children’s Hospital of Zurich (Zurich, Switzerland). The study was approved by the local Swiss ethics committee and registered in national (Swiss National Clinical Trials Portal: KEK 2017-00451) and international (clinicaltrials.gov: NCT03320967) registries.

### 2.1. Participants and Sample Size

All patients, irrespective of their medical condition, between their first day of life and 18th birthday admitted to the intensive care unit of the University Children’s Hospital of Zurich were eligible if the following conditions were met: ability of the caretaker or the adolescent (if ≥14 years of age) to understand verbal and written instructions and informed consent in the local language (German). Exclusion criteria were: impossibility to obtain written informed consent from the caretaker or adolescent (if ≥14 years of age) within 24 h for any reason. Children with syndromes were not excluded.

For the sample size estimation, we took into account the average length of stay in our unit: (median (10–90th percentile)): 1.3 (0.4–9.2) days. Therefore, we defined the study period relevant for the primary outcome (change of copeptin over time, expressed as mean difference between time points) as 48 h, and the sample size was calculated to compensate for patients discharged early. A sample size of 62 × 2.5 × 1.1 = 170 patients was recommended, assuming 80% power, and two-sided 5% significance level with an effect size of 0.36 (with primary analysis, a paired *t*-test for mean difference between baseline and 48 h of *d* = 0.3 ng/mL, approximate *sd* = 0.8 ng/mL [24], 60% drop-out before 48 h, 10% of data below the lower limit of quantification).

### 2.2. Study Design

As soon as written informed consent was obtained from caretakers, the study period began for the included patients. Blood samples for copeptin analysis (EDTA plasma, 600 μL) were planned to be taken from all study patients at specific time points (T0 = admission or study start, if already hospitalized in ICU, T1 ≈ 12 h, T2 ≈ 24 h, T3 ≈ 48 h, T4 ≈ 96 h and T5 ≈ 168 h), processed and stored at −80 °C for later analyses. All methods of blood withdrawal (venous, arterial and capillary) were used. The exact time points for each patient were determined by otherwise medically indicated blood withdrawals to avoid unnecessary pain or catheter manipulation, and therefore, variation in exact blood withdrawal timing was expected. Study blood samples were centrifuged and aliquoted in the internal laboratory of the University Children’s Hospital of Zurich. All plasma tubes were labeled with anonymous barcodes to be analyzed at B∙R∙A∙H∙M∙S GmbH (part of Thermo Fisher Scientific, Waltham, MA, USA) in Hennigsdorf, Germany, via the B∙R∙A∙H∙M∙S Copeptin proAVP KRYPTOR Assay. The lower limit of quantification was 1.23 pmol/L; the upper limit of quantification was 2000 pmol/L. The manufacturer reported for the values between 2 and 4 pmol/L and between 4 and 50 pmol/L intra-assay precision of <15% and <4–8% and inter-assay precision of <18% and <5–10%, respectively.

### 2.3. Variables

The following clinical variables were collected. For baseline assessment: age; sex; weight; Pediatric Index Of Mortality II score (PIM II; percent); main diagnosis (classified as: cardiac birth defects, visceral birth defects of visceral emergencies, respiratory failure, shock or trauma); surgery (classified as: no surgery, cardiac surgery with cardiopulmonary bypass, cardiac surgery without cardiopulmonary bypass, other surgery); cardiopulmonary bypass and its duration (minutes); aortic cross clamp and its duration (minutes); length of respiratory support (invasive, non-invasive, total; hours); length of stay in pediatric intensive care unit (PICU, days), death within 28 days after study inclusion. For the primary endpoint at the time of copeptin blood sampling: systolic and mean blood pressure (mmHg). For secondary endpoints, in addition to the above-mentioned baseline variables: vasopressor-inotrope doses expressed as modified Wernovsky score (Vasoactive-inotropic Score, VIS): dopamine dose (µg/kg/min) + dobutamine dose (µg/kg/min) + 100 × epinephrine dose (µg/kg/min) + 100 × norepinephrine dose (µg/kg/min) + 10,000 × vasopressin dose (U/kg/min) + 10 × milrinone dose (µg/kg/min) [32,33]; total fluids per 24 h (mL/kg/d); steroid dose (mg/kg).

### 2.4. Statistical Endpoint Assessment

The primary analysis was the mean difference in (log-transformed) blood copeptin levels between T0 and T3 (≈48 h), analyzed using a two-sided paired *t*-test. Supplementary analyses also considered the changes in levels between T0 and T1 (≈12 h), T1 and T2 (≈24 h), and T2 and T3 (≈48 h).

The difference in blood copeptin levels between patients with arterial normo- and hypotension between baseline and 48 h after intensive care unit (ICU) admission was analyzed using mixed-effects ANOVA, taking into consideration the repeated measures from each patient with a random intercept effect. Age-specific systolic arterial hypotension was defined according to Appendix A.

Secondary endpoints: The general change in copeptin levels between all consecutive time points (i.e., up to 7 days) was assessed using two-sided paired *t*-tests. The association of the longitudinal trajectory of all obtained blood copeptin levels with clinical covariates was analyzed using mixed-effects linear regression of the log-transformed copeptin values with study time included as both categorical and continuous variable. Residual diagnostics were used to verify the appropriateness of the associated linear modeling assumptions. Univariable models were fitted, followed by adjusted analyses using forward and backward deletion for variable selection with *p* < 0.05 according to the likelihood ratio test, considering all variables from univariate models with *p* < 0.1 as inclusion criteria. The following covariates were investigated: (a) as patient baseline characteristics: age, gender, main diagnosis, PIM II, surgery, type and duration of respiratory support, length of ICU stay and (b) as time-varying variables (time-matched with copeptin sampling): systolic blood pressure (continuous and categorized as hypo-/normotension), mean arterial blood pressure, vasopressor-inotrope use (yes/no) and treatment (VIS: continuous and categorized as 0/>0–10/>10), volume of fluid administration (continuous and categorized as >/≤100 mL/kg/d), steroid use and dose. The interaction of blood-pressure–copeptin association with vasoactive drugs, total fluids and steroid administration were additionally assessed. Continuous covariates were also clinically categorized as follows: age: 0–1 months, 1–12 months, 1–5 years, 5–12 years, ≥12 years; length of ICU stay: >/≤4 days; respiratory support (invasive, non-invasive, total): ≤24 h, 24–48 h, 48–168 h, >168 h; total fluids >/≤100 mL/kg/day; cardiopulmonary bypass (if performed): <60 min, 60–120 min, 120–180 min, >180 min; aortic cross clamping (if performed): <60 min, 60–120 min, >120 min.

Mechanistic modeling of the time course was initially planned but was not performed, as no typical kinetic pattern was observed. Statistical significance was set to *p* < 0.05, unless stated otherwise.

## 3. Results

### 3.1. Patients

Between December 2017 and June 2019, 170 patients were included in this study. One patient was discharged from the ICU before the first blood sample could be taken. For one patient, the caretakers signed the consent before a planned surgery, but the patient did not need postoperative intensive care unit admission. Out of the remaining 168 patients, 164 patients (characteristics: Table 1) had paired (time-matched) copeptin and systolic blood pressure samples collected (*n* = 583 samples: *n* = 164 at T0 = ICU admission/study start, and *n* = 150/106/80/51/32 at median actual time (interquartile range) into the study of T1 = 14 h (11, 16); T2 = 37 h (34, 40); T3 = 61 h (58, 63); T4 = 108 h (97, 110); T5 = 179 h (168,182). Overall (for all time points together), the median copeptin was 46 pmol/L (20.4–114) (range: 2.4–1552.6 pmol/L). In total, 83 (50.6%) patients experienced at least one period of hypotension, corresponding to 211 out of 583 (35.2%) matched measurements with systolic blood pressure values and copeptin values obtained simultaneously. The changes in copeptin over time overall and stratified by hypotension are shown in Figure 1A,B and Figure 2, respectively.

### 3.2. Primary Endpoint: Change of Copeptin over 61 h and Association with Arterial Hypotension

Given the observed median measurement time for the T3 time point, we adjusted the primary endpoint to consider the average change over the first 61 h (rather than the planned 48 h). For the primary analysis, the description of copeptin change between baseline and T3, there was no difference (mean difference on log scale: −0.13, corresponding to mean individual change of −12%, 95%CI: −33 to16%, *p* = 0.36). Mean copeptin levels were higher in patients with hypotension (estimate: 0.39, corresponding to 47% or 1.47-fold increase, 95%CI: 1.07–2.01-fold increase, one-way ANOVA from the mixed-effects model, *p* = 0.01); mean copeptin levels at T0 were 43.5 pmol/L (geometric mean) in normotensive patients (=reference). We found eight patients with low copeptin values during arterial hypotension (individual case presentations available in Appendix A).

Furthermore, between T0 and T1 (difference on log-scale: +0.19, *p* = 0.84), and between T2 and T3 (−0.12, *p* = 0.19), a difference was found. Individual copeptin levels significantly decreased between T1 and T2: −0.52, corresponding to a mean individual change by −41%, *p* < 0.001.

### 3.3. Secondary Endpoints

#### 3.3.1. Change of Blood Copeptin Levels between Consecutive Time Points up to 7 Days

When considering the change in copeptin levels between all consecutive time points, individual copeptin levels significantly decreased between T3 and T4 (−0.32, corresponding to −27%, *p* = 0.01). Between the other time points, no significant change was observed (i.e., T4 and T5, −0.11, *p* = 0.582).

#### 3.3.2. Association of Blood Copeptin Levels with Clinical Covariates in Linear Mixed-Effect Model 

The results of the mixed-effects linear regression of log-transformed copeptin values are shown in Table 2. In the univariate analysis, the vasopressor-inotrope treatment showed the strongest association with log-transformed copeptin levels (linear positive association with log-transformed VIS). Significant associations (ordered by *p*-value of likelihood ratio test) were also found for surgery/type of surgery (lowest copeptin in patients without surgery, increased levels in patients after non-CBP cardiac or non-cardiac surgery, highest levels after CPB surgery), study time (highest levels at T1, thereafter declining, with lower levels at T4 and T5 compared to T0, respectively), main diagnosis (levels in patients with cardiac birth defects >> shock ≥ other ≥ respiratory failure; differences for trauma and visceral birth defects not significant), age (highest levels in infants 1–12 months, lowest levels in children 5–12 years, no significant difference between neonates, children 1–5 years and adolescents), systolic blood pressure (negative linear association: decrease by −8.4% for each 10 mmHg increase, *p* = 0.003, versus categorization: +15% in hypo- versus normotension, *p* = 0.14); association with mean arterial pressure: *p* = 0.034), steroid use (positive linear association with dose), total fluid volume (lower copeptin in patients with high fluid administration > 100 mL/kg/d compared to those receiving ≤100 mL/kg/d), length of stay (lower levels in patients staying ≤4 days compared to >4 days), invasive mechanical ventilation (positive linear association with duration; no significant association with non-invasive ventilation or total duration of invasive/non-invasive ventilation) and PIM II (log-transformed: positive linear association). No significant associations were found for the other covariates, including the interaction terms.

In the adjusted analysis, only the vasopressor-inotrope treatment (14% increased copeptin by each 2.7-fold increase in VIS, Figure 3A), age (65% higher copeptin in infants 1–12 months compared to neonates; older age groups: no significant difference compared to neonates, Figure 3B), study time (20% higher levels at T1 and 30% lower levels at T4 compared to T0, respectively; other time points: not significantly different) and surgery/type of surgery remained significant (2.2-fold higher copeptin in patients with CPB surgery compared to patients without surgery; non-CPB surgery: estimated 54% higher levels compared to patients without surgery, not significant, Figure 3C).

## 4. Discussion

This prospective study evaluated longitudinal blood copeptin in critically ill children and its association with arterial normo- and hypotension. We hypothesized that arterial hypotension would be associated with low or normal copeptin concentrations < 10 pmol/L, according to the published copeptin values for healthy individuals, providing a hint for absolute or relative AVP deficiency. While our first hypothesis was not confirmed, we detected increased copeptin levels in arterial hypotension. Moreover, we found blood copeptin levels to be associated with severity of disease markers, i.e., vasoactive support, length of mechanical ventilation, length of ICU stay, PIM II score, steroid use and intravenous fluid administration. Thus, our second hypothesis was confirmed and underscores the physiologic AVP release in life-threatening hypotension and stress. Accordingly, copeptin might serve as a surrogate marker for severity of disease in critically ill children.

### 4.1. Low Copeptin as Surrogate for AVP Deficiency

Children in intensive care units frequently suffer periods of arterial hypotension. The reasons are mainly circulatory insufficiency in shock states and side effects of sedation and/or invasive mechanical ventilation. Arterial hypotension and shock are generally treated with intravenous fluids, vasopressors and inotropes, and steroids, such as hydrocortisone [1,34,35]. Mastropietro et al. raised the question of whether endogenous relative or absolute AVP deficiency contributes to vasodilation, leading to arterial hypotension and shock [19]. The authors suggested the measurement of blood copeptin as a stable surrogate marker for AVP to guide exogenous AVP administration [18]. In the present study, blood copeptin was elevated in hypotensive patients, in particular during the first 48 h (1.47-fold increase compared to arterial normotension). This was not unexpected, as arterial hypotension and stress are among the main drivers of increased AVP release from the neurohypophysis, alongside copeptin [36,37,38]. The elevation of copeptin concentrations represents the regular physiologic reaction to these stressors. Nonetheless, clinicians are sometimes faced with refractory arterial hypotension, followed by a breakthrough rise in arterial blood pressure after administration of vasopressin. These impressive clinical courses point toward absolute or relative AVP deficiency. In our study, we only identified 8 hypotensive patients with copeptin concentrations <10 pmol/L (out of 34). These eight children were in stable clinical conditions at the time of measurement. Consequently, this study does not promote the measurement of copeptin to tailor vasopressin therapy individually according to copeptin values. There might be several reasons for this. First, there is no internationally valid normal value for blood copeptin concentrations for children or adults. Using published studies, including healthy controls, we arbitrarily set the cut-off for normal blood copeptin at 10 pmol/L [27,39,40,41]. It may well be that this value was set too low and that the baseline copeptin concentration for children admitted to the intensive care unit is generally higher. Mastropietro et al. used a copeptin cut-off of 1.12 ng/mL [18] to define AVP deficiency in a postoperative cohort. This would be equivalent to a copeptin value of 206 pmol/L, a very high value compared to the median copeptin concentration of 46 pmol/L found in this study. Second, we did not measure the AVP concentrations for direct comparison to blood copeptin. Hence, we were not able to directly define AVP deficiency and the corresponding copeptin values. Third, AVP and copeptin are synthesized in an equimolar 1:1 ratio, but the decay kinetics and the underlying mechanism may be different [42]. It may well be that the phases of arterial hypotension occurred without an adequate rise of AVP, but the copeptin concentration was still elevated from the former AVP release.

### 4.2. Course of Copeptin in Normotensive and Hypotensive Patients

Arginine-vasopressin plays a key role in body water and electrolyte homeostasis. Hypovolemia and low blood pressure are detected by vascular volume- and baroreceptors and drive AVP and copeptin release in the posterior pituitary gland [43,44]. In the present study, the median copeptin concentration in all patients (hypotensive and normotensive) was calculated with a median of 47 pmol/L, five-fold higher compared to the published normal values in healthy children (<10 pmol/L) [27,39,40,41]. The median copeptin concentrations remained elevated in all patients over 168 h, a finding that mirrors increased AVP and copeptin values measured over 120 h in pediatric septic shock patients by Lee et al. [24]. In the case of arterial hypotension, copeptin concentrations showed a tendency toward elevated (+15%) blood copeptin over the 168 h study period and a significant negative correlation with blood pressure (−8.4% for each 10 mmHg increase). The categorization by normotension/hypotension achieved statistical significance during the first 61 h (+47% higher levels during hypo- versus normotension), most likely representing an adequate physiologic reaction to arterial hypotension, hypovolemia and/or stress. As we included many postsurgical patients (84%), this time period may represent postoperative fluid restriction, vasodilation caused by postsurgical systemic inflammatory response syndrome (SIRS) and sedatives. Hypovolemia and arterial hypotension might be the main stimuli for AVP and copeptin release, potentially alongside peri- or postoperative pain and stress [43].

### 4.3. Copeptin as Surrogate Marker for Severity of Disease

In recent years, blood copeptin has been recognized as a reliable surrogate marker for severity of disease and adverse outcomes. In adults, it has been shown to predict adverse outcomes in ischemic [45] and hemorrhagic stroke [46], heart failure [47], community-acquired pneumonia [48], COVID-19 [49] and many more [50,51,52]. In children, copeptin predicted an adverse outcome in community-acquired pneumonia [25,26], congenital heart disease and pulmonary arterial hypertension [27], perinatal asphyxia [28], traumatic brain injury [31] and severity of disease, as well as mortality, in PICU [53]. In the present study, high copeptin levels were associated with the length of ICU stay, length of mechanical ventilation, dose of vasoactive and inotropic support, additional fluids exceeding 100 mL/kg/d, PIM II Score and steroid administration in univariate analysis, while only vasoactive support remained significant in the adjusted (multivariate) analysis. These results support the usefulness of copeptin as a surrogate for severity of disease shown in the previous studies mentioned above. Copeptin was further associated with age, being highest in the infant group of 1–12-month-olds (65% higher compared to neonates). It is important to recognize that many infants were postsurgical and post-CPB patients. Thus, their copeptin concentrations might have been affected by the surgical intervention and not by age-specific physiology. However, age retained its significant influence after accounting for surgery. Further, baseline copeptin values were shown to be gender dependent in past studies [23,54,55], but in this study, no gender dependency was found. The reason might be that the strong stimuli for AVP and copeptin release during medical and postsurgical intensive care treatment overruled the gender disparity. Interestingly, copeptin was associated with cardiopulmonary bypass (2.2-fold higher levels observed compared to patients without surgery) and performance of the aortic cross clamping per se but not with their duration. These procedures seem to initiate the relevant stimulus, which is not aggravated by a longer time period on CPB or by a longer aortic cross clamp time. This is in line with two studies performed by Mastropietro et al., which did not show an influence of CPB or aortic cross clamp duration neither on AVP nor on copeptin release [18,19].

### 4.4. Limitations

This study has important limitations that need consideration. First, we did not include a healthy control group. This would have allowed the comparison of copeptin values from sick and healthy patients. Second, many studies assessed the predictive value of copeptin for mortality. In this cohort, only one patient died within 28 days, and the association with mortality could not be evaluated with a sound statistical approach. This one patient was a 7-week-old male patient with Trisomy 21 and atrioventricular septum defect who was put on extracorporeal membrane oxygenation (ECMO) postoperatively. He died on the 15th postoperative day on ECMO due to acute bleeding complications following a surgical revision. His copeptin values were 33.3/12.6/16.6/10.7/28.7/17.7 pmol/L at T0–T5. These were well within the range of the other patients and did not point toward a non-favorable outcome. Third, we used PIM II score as a surrogate for severity of disease. The inclusion of PIM II score was based on availability in the study hospital. We extracted the actual PIM II score from patients’ medical records and did not calculate it because of the preference for real-world data. There might be newer scores [56] with higher accuracy. Lastly, larger cohorts with follow-ups should be evaluated for associations with long-term outcomes.

## 5. Conclusions

Median copeptin concentrations were significantly elevated in critically ill children when compared to published copeptin values of healthy children. In addition, this study did not evidence a correlation between arterial hypotension and inadequate low copeptin concentrations. Therefore, copeptin’s potential role in guiding exogenous vasopressin administration seems to be limited in neonatal and pediatric intensive care patients. Importantly, copeptin was positively associated with clinical conditions reflecting disease severity and might be useful as a marker to predict short-term outcomes.

## Figures and Tables

**Figure 1 children-09-00794-f001:**
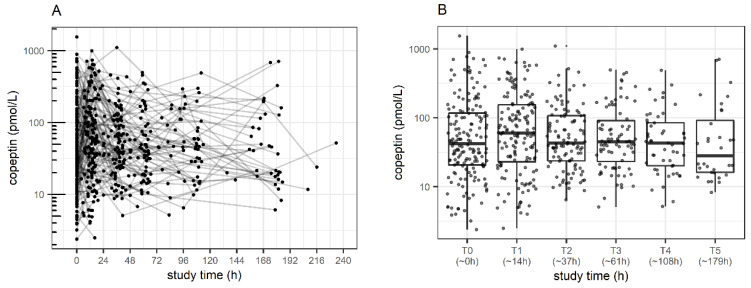
Blood copeptin levels over the study period of 168 h in critically ill children: (**A**) copeptin concentrations with individual trajectories over time; (**B**) blood copeptin concentrations over time presented as box plots. Boxes show the interquartile range (IQR). Solid lines are the median, 25th and 75th quantile, and whiskers equal to 25th quantile −1.5 IQR and 75th quantile +1.5 IQR. Corresponding time points: T0 = study start/T1 = 14 h (median, interquartile range (11, 16 h)); T2 = 37 h (34, 40); T3 = 61 h (58, 63); T4 = 108 h (97, 110); T5 = 179 h (168, 182).

**Figure 2 children-09-00794-f002:**
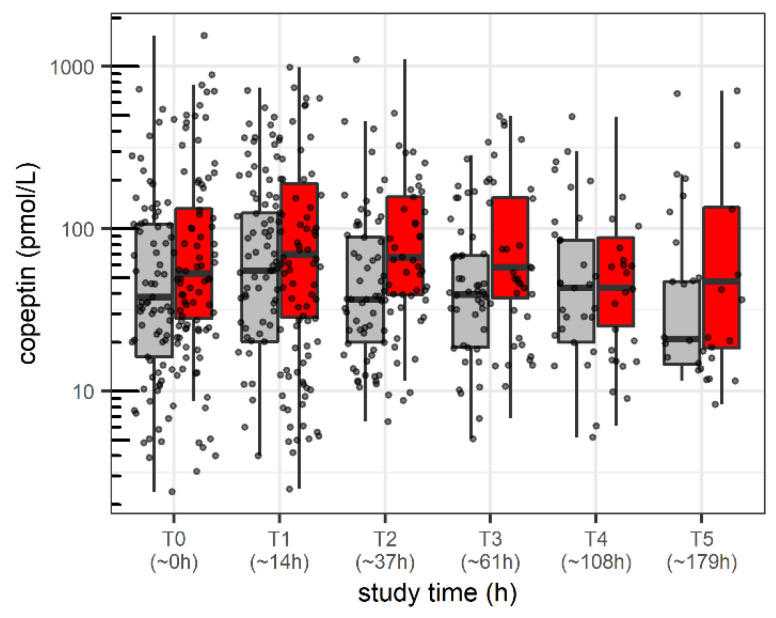
Blood copeptin levels over the study period stratified by presence of arterial hypotension, red boxes; arterial normotension, gray boxes. Boxes represent the interquartile range (IQR). Solid lines are the median, 25th and 75th quantile, and whiskers equal to 25th quantile −1.5 IQR and 75th quantile +1.5 IQR. Corresponding time points: T0 = study start; T1 = 14 h (median, interquartile range (11–16 h)); T2 = 37 h (34–40 h); T3 = 61 h (58–63 h); T4 = 108 h (97–110 h); T5 = 179 h (168–182 h).

**Figure 3 children-09-00794-f003:**
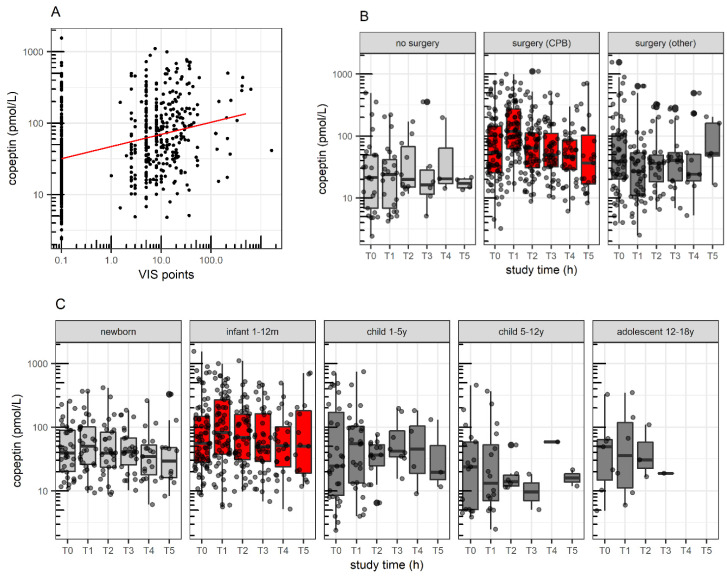
Blood copeptin levels stratified by variables that remained significantly associated in the adjusted mixed-effect regression. (**A**) Blood copeptin levels versus vasopressor-inotrope treatment (VIS points on log-scale, VIS = 0 is plotted at 0.1). (**B**) Blood copeptin levels over the study period stratified by surgery, red boxes = copeptin levels in patients after cardiopulmonary bypass (CPB) surgery were significantly higher than in patients without surgery (=reference). (**C**) Blood copeptin levels over the study period stratified by age, red boxes = copeptin levels in infants 1–12 month of age were significantly higher than in neonates (=reference). Boxes represent the interquartile range (IQR). Solid lines are the median, 25th and 75th quantile, and whiskers equal to 25th quantile −1.5 IQR and 75th quantile +1.5 IQR. Corresponding time points: T0 = study start/T1 = 14 h (median, interquartile range (11–16 h))/T2 = 37 h (34–40 h)/T3 = 61 h (58–63 h)/T4 = 108 h (97–110 h)/T5 = 179 h (168–182 h).

**Table 1 children-09-00794-t001:** Baseline characteristics of the 164 patients included in the analysis.

Patient Characteristics	Value
Age (years)	0.5 [0.1, 2.9]
Newborn (0–1 month)	35 (21.3)
Preterm (<37 weeks gestational age)	3 (1.8)
Infant (1–12 month)	75 (45.7)
Preschool (1–5 years)	30 (18.3)
Child (5–12 years)	17 (10.4)
Adolescent (12–18 years)	7 (4.3)
Weight (kg)	6.1 [4.0, 13.3]
Female gender	68 (41.5)
Main diagnosis	
Cardiac birth defects	99 (60.4)
Visceral birth defects/Visceral emergencies	16 (9.8)
Respiratory failure	10 (6.1)
Shock	7 (4.3)
Trauma	1 (0.6)
Other *	31 (18.9)
Length of ICU stay (days)	4.0 [1.0, 10.5]
Respiratory support (any)	136 (82.9)
Length of respiratory support, total (h)	25.7 [3.3, 114.1]
Invasive mechanical ventilation	133 (81.1)
Length of invasive mechanical ventilation (h)	20.5 [2.8, 98.4]
Vasoactive-inotrope medication (at any time)	103 (61.7)
Surgery	139 (84.7)
Cardiopulmonary bypass (CPB)	80 (48.8)
CPB duration (min)	140.5 [97.8, 212.5]
CPB duration (≤60 min)	2 (2.5)
CPB duration (60–120 min)	26 (32.5)
CPB duration (120–180 min)	23 (28.8)
CPB duration (≥180 min)	29 (36.3)
Aortic cross clamp time (min) (*n* = 76)	83.0 [58.0, 111.2]
Aortic cross clamp time (≤60 min)	22 (29.0)
Aortic cross clamp time (60–120 min)	39 (51.3)
Aortic cross clamp time (≥120 min)	15 (19.7)
Death within 28 days	1 (0.6)

Data are presented as median [interquartile range] for continuous variables or *n* (%) for categorical variables, respectively. * The following diagnoses were grouped to “other” for low individual numbers: acute endocarditis, arterial hypertonic crisis, brain tumor, cerebral palsy, chronic rhinosinusitis, congenital atrioventricular block III°, epipharyngeal bleeding, facial hypoplasia, fever of unknown origin, unstable sternum, intracerebral hemorrhage, meningocele, mitochondriopathy, moya-moya disease, myocarditis, pulmonary hypertension, renal failure, scaphocephaly, scoliosis, subcutaneous cerebral fluid collection, vasculitis, venous malformation.

**Table 2 children-09-00794-t002:** Association of blood copeptin levels with clinical covariates in linear mixed-effect model (*p*-value: likelihood ratio test).

Variable	*p*-Value	Adjusted *p*-Value
Vasoactive-inotropic Score (VIS) #, log-transformed (linear)	<0.001	<0.001
Surgery (no surgery, cardiac surgery with CPB, other surgery)	<0.001	<0.001
Study time point (categorical)	<0.001	0.002
Main diagnosis	<0.001	-
Age (0–1 m, 1–12 m, 1–5 y, 5–12 y, ≥12 y)	<0.001	0.001
Systolic blood pressure (linear)	0.003	-
Steroid dose (linear)	0.007	-
Total fluid (0/≥/<100 mL/kg/day)	0.011	-
Duration of invasive mechanical ventilation (linear)	0.014	
Length of stay (>/≤ 4 days)	0.021	-
PIM II, log-transformed (linear)	0.034	-
Normo-versus hypotension	0.14	-
Gender	0.63	-
Duration of CPB surgery (≤60 min/60–120/120–180/≥180 min) *	0.18	-
Aortic cross clamp time (linear) *	0.52	-

# VIS points equaling 0 were set to 0.1. * subset of patients with CPB surgery.

## Data Availability

All data generated or analyzed during this study are included in this article and in Appendix A. Further enquiries can be directed to the corresponding author.

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
