# Peer review of "Copeptin Release in Arterial Hypotension and Its Association with Severity of Disease in Critically Ill Children"

_children, 2022, doi:10.3390/children9060794_

Round 1

Reviewer 1 Report

Very interesting and well presented research. I have no suggestions, except one: In Table 1, in parentheses, indicate the percentages (%) to make it clearer

Author Response

Request by Reviewer #1

Very interesting and well presented research. I have no suggestions, except one: In Table 1, in parentheses, indicate the percentages (%) to make it clearer.

Authors` response: We are not sure, if we understood the reviewer comment in the right way, as “Data are presented as median [interquartile range] for continuous variables or n (%) for categorical variables, respectively” was already specifically stated in the table`s caption. Thus, we think that the requested information is already there.

Reviewer 2 Report

Thank you for presenting an important topic on hypotension in sick pediatric population 

1-Please clarify that the infants included in the study are all term infants or any percentage of patients are premature as that population can have difference in hormonal levels( AVP levels ) ; since a substantial part of your study population is less than one year of age .

2- Inclusion criterion needs to be defined in more  details 

3-Please add supplement for age specific hypotension criterion , did authors used mean blood pressures for hypotension .

4-Did authors collect data on genetic tests - chromosomes /microarrays /Whole exome / did they exclude syndromic infants ?

5-Table 1 , what does other means ?

Author Response

Requests by Reviewer #2

Thank you for presenting an important topic on hypotension in sick pediatric population 

1-Please clarify that the infants included in the study are all term infants or any percentage of patients are premature as that population can have difference in hormonal levels (AVP levels) ; since a substantial part of your study population is less than one year of age.

Authors` response: The authors are grateful for the reviewer`s comment, this information was indeed missing so far. There were three preterm children included in the study cohort, born at 34 weeks + 3 days, 33 weeks + 5 days, and 35 weeks + 0 days of gestational age. These children represent 1.8 % of the study cohort (n=164) and are not deemed to introduce relevant bias. However, we included this information into Table 1.

2- Inclusion criterion needs to be defined in more details 

 Authors` response: We thank the reviewer for this input. Both the inclusion and exclusion criteria were defined as follows:

Inclusion criteria:

  • Age: first day of life until 18th birthday.
  • Ability of the care taker or the adolescent (if≥14 years of age) to understand verbal and written instructions and informed consent in German.

Exclusion criteria:

  • Care taker or adolescent (if ≥14 years of age) unwilling to give written informed consent.
  • Care taker or adolescent (if ≥14 years of age) not understanding German and without a family member able to translate.
  • Adolescent (if ≥14 years of age) unwilling to give written informed consent following sedation < 24 hours.
  • Care takers of long-term sedated (>24 hours) adolescents (if ≥14 years of age) unwilling to give written informed consent or not present within 24 hours.

Importantly, the in- and exclusion criteria were prospectively published on https://clinicaltrials.gov/ct2/show/NCT03320967. Full study protocol containing the original in- and exclusion criteria can be seen there as well.

We would like to point out that we investigated copeptin course in our broad NICU/PICU population, i.e. in a rather non-selective cohort. However, considering the reviewer’s comment we now included a statement on the in-/exclusion of syndromic infants.

 3-Please add supplement for age specific hypotension criterion , did authors used mean blood pressures for hypotension.

Authors` response: During study planning, we evaluated systolic and mean arterial pressures as possible thresholds for defining arterial hypotension. Normal values for systolic blood pressure in children are better defined than mean arterial pressures. Therefore, we decided to use systolic arterial pressure to define arterial hypotension. This was stated in the first submitted version of the article in section “2.4. Statistical endpoint assessment” and especially in Table S1 including references used to generate hypotension thresholds.

However, in the data analysis process we compared the associations between copeptin and systolic as well as mean arterial blood pressure: interestingly, systolic blood pressure showed a stronger association with copeptin values (p=0.003) than mean arterial pressure (p=0.034). This was commented already in section 3.2.2 of the first submitted version of the article. We did not change the text in the updated version.

 4-Did authors collect data on genetic tests - chromosomes /microarrays /Whole exome / did they exclude syndromic infants?

Authors` response: No, we were not able to collect genetic material for chromosomes, microarrays, etc. as this would require a different study approach and especially, at least in Switzerland, a separate parental consent. Children with syndromes were not excluded and this was entered as statement into the section “2.1. Participants and sample size” in the revised version.

 5-Table 1, what does other means?

Authors` response: Indeed, this information was missing so far, we are grateful for the editor`s comment. A phrase was inserted to the caption of table 1 explaining “other”.